

# Investigating *Escherichia coli* habitat transition from sediments to water in tropical urban lakes

Boyu Liu[1], Choon Weng Lee[1,2], Chui Wei Bong[1,2] and Ai-Jun Wang[3,4]

[1] Laboratory of Microbial Ecology, Institute of Biological Science, Faculty of Science, Universiti Malaya, Kuala Lumpur, Malaysia
[2] Institute of Ocean and Earth Sciences, Universiti Malaya, Kuala Lumpur, Malaysia
[3] Laboratory of Coastal and Marine Geology, Third Institute of Oceanography, Ministry of Natural Resources, Xiamen, Fujian, China
[4] Fujian Provincial Key Laboratory of Marine Physical and Geological Processes, Xiamen, Fujian, China

Corresponding authors
Choon Weng Lee, lee@um.edu.my
Ai-Jun Wang, wangaijun@tio.org.cn

## ABSTRACT

**Background**. *Escherichia coli* is a commonly used faecal indicator bacterium to assess the level of faecal contamination in aquatic habitats. However, extensive studies have reported that sediment acts as a natural reservoir of *E. coli* in the extraintestinal environment. *E. coli* can be released from the sediment, and this may lead to overestimating the level of faecal contamination during water quality surveillance. Thus, we aimed to investigate the effects of *E. coli* habitat transition from sediment to water on its abundance in the water column.

**Methods**. This study enumerated the abundance of *E. coli* in the water and sediment at five urban lakes in the Kuala Lumpur-Petaling Jaya area, state of Selangor, Malaysia. We developed a novel method for measuring habitat transition rate of sediment *E. coli* to the water column, and evaluated the effects of habitat transition on *E. coli* abundance in the water column after accounting for its decay in the water column.

**Results**. The abundance of *E. coli* in the sediment ranged from below detection to 12,000 cfu g$^{-1}$, and was about one order higher than in the water column (1 to 2,300 cfu mL$^{-1}$). The habitat transition rates ranged from 0.03 to 0.41 h$^{-1}$. In contrast, the *E. coli* decay rates ranged from 0.02 to 0.16 h$^{-1}$. In most cases (>80%), the habitat transition rates were higher than the decay rates in our study.

**Discussion**. Our study provided a possible explanation for the persistence of *E. coli* in tropical lakes. To the best of our knowledge, this is the first quantitative study on habitat transition of *E. coli* from sediments to water column.

## INTRODUCTION

Faecal indicator bacteria (FIB) are a group of bacteria used to evaluate water faecal contamination. Ideally, FIB should be of faecal origin only and not grow in the extraintestinal environment (*Rochelle-Newall et al., 2015*). Furthermore, the abundance of FIB should correlate with the presence of faecal contamination-related pathogen.

Compared with direct detection of these pathogens, FIB are more abundant in the water and thus easier to detect (*Tortora, Funke & Case, 2013*).

Globally, *Escherichia coli* has been used as a FIB since the last century (*USEPA, 1986*). Due to its wide application, extensive studies have been done on its survival in water. The survival of *E. coli* in aquatic habitats is affected by both biotic and abiotic factors (*Jang et al., 2017*). For example, biotic factors include biofilm formation and the presence of other microorganisms (*Korajkic et al., 2014*; *Stocker et al., 2019*), whereas abiotic factors include temperature, pH, salinity, sunlight and nutrient availability (*Petersen & Hubbart, 2020*; *Moon et al., 2023*). Therefore, seasonal variations with changes in temperature, precipitation and anthropogenic activity could also affect *E. coli* abundance and their survival. *An, Kampbell & Peter Breidenbach (2002)* reported lowest *E. coli* density in summer and attributed this to the lower loading of faecal material, more vigorous grazing, and poor survival of *E. coli* in warm water. However, *Durham et al. (2016)* reported highest *E. coli* abundance in summer, suggesting that site-specific factors are also relevant.

Nevertheless, there remain doubts about *E. coli*'s reliability as a FIB, as studies have revealed that sediments are an environmental reservoir of *E. coli* in freshwater habitats (*Ishii et al., 2006*; *Ishii & Sadowsky, 2008*; *Cho et al., 2010*; *Garzio-Hadzick et al., 2010*; *Tymensen et al., 2015*; *Fluke, González-Pinzón & Thomson, 2019*). Relative to the water column, sediments generally have higher nutrient levels, lower dissolved oxygen, and lower UV intensity, which helps *E. coli* survive in sediments (*Jamieson et al., 2005*; *Koirala et al., 2008*; *Lorke & MacIntyre, 2009*; *Rochelle-Newall et al., 2015*). Studies have also reported that habitat transition of sediment *E. coli* to the water column, increases *E. coli* abundance in the water; for example, during resuspension of sediment by mechanical effects like precipitation or water flow (*Whitman, Nevers & Byappanahalli, 2006*; *Cho et al., 2010*; *Abia et al., 2017*). Apart from resuspension, habitat transition should theoretically also occur as *E. coli* grows in the sediments (*Ishii et al., 2006*). For instance, *E. coli* that thrives on sediment biofilms can be released into the water due to biofilm sloughing (*Mackowiak et al., 2018*).

Previous habitat transition studies focused more on sediment resuspension induced by mechanical effects. These mechanical effects included anthropogenic vessel activity and precipitation caused by seasonal variation. *An, Kampbell & Peter Breidenbach (2002)* revealed the resuspension of sediment caused by motorboat leads to water quality deterioration. Precipitation can also cause the resuspension of sediment, causing *E. coli* habitat transition from sediment to the upper water column (*Li, Filippelli & Wang, 2023*). However, increase in *E. coli* due to sediment resuspension will quickly return to pre-resuspension concentrations (*Whitman, Nevers & Byappanahalli, 2006*; *Abia et al., 2017*). Besides that, our literature review revealed no report that measured sediment *E. coli* habitat transition rates to the overlying waters.

As the habitat transition rate could be an important process that contributes to *E. coli* prevalence in the waters, we designed experiments to measure the habitat transition rate of *E. coli* in sediment samples from lakes. In this study, five tropical urban lake waters were selected, as lake waters are generally more static and have less sediment resuspension (*Lim et al., 2018*; *Bong et al., 2020*). The absence of mechanical effects in the lake waters will

help clarify the role of *E. coli* habitat transition. Since the abundance of *E. coli* in the upper water column is also affected by its growth or decay, we concurrently carried out habitat transition experiments with size-fractionation decay experiments according to *Lee et al. (2011)*. Our results helped shed light on the possible reasons for the persistence of *E. coli* in urban tropical lakes as shown earlier by *Wong et al. (2022)*, and could help improve the current water surveillance strategies.

## MATERIALS & METHODS

### Study sites and environmental variables

A total of 35 water and 21 sediment samples were collected regularly at five independent urban lakes (Tasik Varsiti, Tasik Taman Jaya, Tasik Aman, Tasik Kelana and Tasik Central Park Bandar Utama), located 2–7 km between each other in the Klang Valley, Peninsular Malaysia, from May 2022 until November 2022 (Fig. 1). The sampling dates with respective coordinates and experiments conducted for each sampling are listed in Table S1. To avoid effects of precipitation, sampling was carried out when there was no rain. Surface water samples ($\approx 0.1$ m) were collected using autoclaved bottles (121 °C at 15 psi for 15 min) whereas surface sediment samples ($\approx$ three cm depth) were taken with a shovel and collected using UV sterilized (at 245 nm wavelength for 20 min, intensity 550 $\mu$W cm$^{-2}$) plastic zip lock bags. All samples were transferred on ice to the laboratory within 3 h for further analysis.

A conductivity probe (YSI Pro 30, Yellow Springs, OH, USA) and a pH meter (Hach HQ11d, Loveland, CO, USA) were used to measure *in-situ* water temperature and pH, respectively. For dissolved oxygen (DO), water samples were collected with DO bottles in triplicates, and fixed with manganese chloride and alkaline iodide solution, before titration with sodium thiosulphate solution according to Winkler's method (*Grasshoff, Kremling & Ehrhardt, 1999*). Total suspended solids (TSS) was determined by filtering a known volume of water sample through a pre-combusted glass fibre filter (GF/F) (Sartorius, Goettingen, Germany) and measuring the weight increase after drying at 70 °C for a week. Particulate organic matter (POM) was determined by the weight loss after combustion at 500 °C for 2 h (HYSC MF-05, Seoul, Korea). Chlorophyll *a* (Chl *a*) was also concentrated on the GF/F filter and extracted with 90% (v/v) ice-cold acetone at $-20$ °C overnight. Chl *a* concentration was then measured *via* a spectrofluorometer (PerkinElmer LS55, Waltham, MA, USA) (*Parsons, Maita & Lalli, 1984*). The filtrate from the filtration was kept frozen until the determination of ammonium ($NH_4$) and phosphate ($PO_4$). These dissolved inorganic nutrients were determined on a spectrophotometer (Hitachi U-1900, Tokyo, Japan) *via* methods described by *Parsons, Maita & Lalli (1984)*.

The sediment sample collected was dried in a freeze dryer (Labconco FreeZone 6 Liter, Kansas City, MO, USA). For sediment particle sizing, about 10 cm$^3$ of dried sediments were mixed with distilled water until a final volume of 40 mL. Then 10 mL of sodium hexametaphosphate (20% final concentration) was added to disperse the sediment particles (*Mil-Homens et al., 2006*). The prepared sample was then homogenized and left overnight before analysis with the Beckman Coulter LS230 Particle Size Analyzer (Brea, CA, USA).
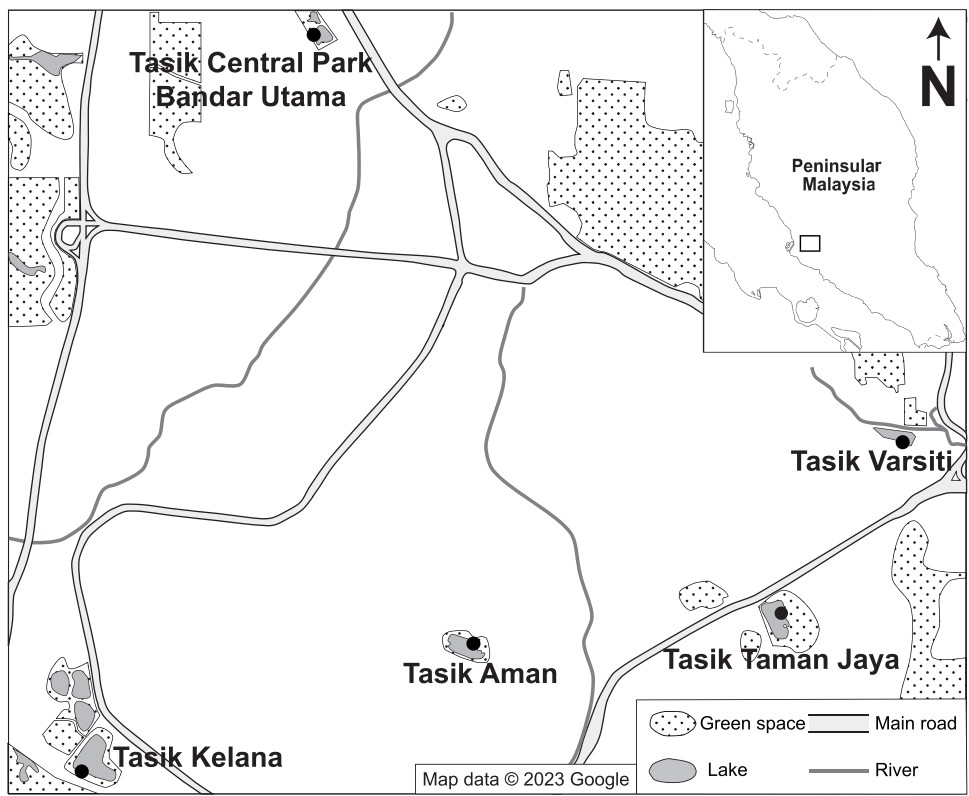

**Figure 1** **Map of the location of sampling stations.** Map showing the location of the tropical urban lakes (Tasik Varsiti, Tasik Taman Jaya, Tasik Aman, Tasik Kelana and Tasik Central Park Bandar Utama) sampled at the Kuala Lumpur–Petaling Jaya area, Malaysia. Map data ©2023 Google (*Google, 2023*).

For sediment organic matter content, the freeze-dried sediment was combusted at 500 °C for 3 h, and the organic matter content was measured *via* the loss on ignition method (*Heiri, Lotter & Lemcke, 2001*).

## Enumeration of coliform and *E. coli* in water and sediment samples

For water samples, both coliform and *E. coli* were measured whereas for sediment samples, only *E. coli* was measured. The additional coliform measurement in the water samples helped in the classification of the lake waters according to the National Water Quality Standards for Malaysia (*Department of Environment, 2008*). Membrane filter technique (MFT) was used to enumerate both coliform and *E. coli* in water where a known volume of water sample (0.01 mL to 10 mL) was filtered through a sterile 47 mm diameter, 0.45 μm pore-size nitrocellulose membrane filter (Millipore, Burlington, MA, USA). For volumes <1.0 mL, the filtration vessel was filled with 5 mL sterile saline (0.85% sodium chloride (NaCl) final concentration) before addition of sample. After filtration, the membrane filter was placed on the CHROMagar™ ECC agar (CHROMagar, Paris, France) and incubated at 37 °C for 24 h. All blue and mauve-coloured colonies were counted as total coliform, whereas only blue colonies were counted as *E. coli* (*Chromagar, 2019*).

For sediment samples, 2 g of fresh sediment sample was mixed with 18 mL of sterile saline and then sonicated for 50 s with an ultrasonicator (220 W, 2 mm probe; SASTEC ST-JY98-IIIN, Subang Jaya, Malaysia) (*Epstein & Rossel, 1995*). After allowing the mixture to settle for 10 min, the suspension was pipetted and used as inoculum in the MFT described above. The membrane filter was then placed on m-TEC agar (Sigma-Aldrich, Burlington, MA, USA) and incubated at 44.5 °C for 24 h. Purple- or magenta-coloured colonies were counted as *E. coli* (*Merck KGaA, 2018*).

## Measuring *E. coli* decay or growth rates

Using the size fractionation method, the water sample was divided into three size fractions: total or unfiltered, <20 µm and <0.2 µm fractions (*Lee et al., 2011*). The <20 µm fraction was collected after filtration through a nylon net with a 20 µm mesh opening size, whereas the <0.2 µm fraction was collected after filtration with a 0.2 µm pore-size membrane filter (Millipore GTTP filter, Burlington, MA, USA).

As *E. coli* counts in the water was sometimes too low, we used a laboratory strain of *E. coli* (isolated from Tasik Varsiti) for the decay or growth experiment. A fresh *E. coli* culture was adjusted to 0.5 McFarland standard (about $1.5 \times 10^8$ cfu mL$^{-1}$) before further serial dilution to $10^5$ cfu mL$^{-1}$. About 198 mL of each size fraction was then inoculated with 2 mL of $10^5$ cfu mL$^{-1}$ *E. coli* culture for a final concentration of about $10^3$ cfu mL$^{-1}$. Inoculated size fractions were then incubated at 30 °C for 24 h in the dark. The abundance of *E. coli* was determined as cfu mL$^{-1}$ every 6 h through MFT on m-TEC agar. The cfu data was then transformed *via* natural logarithm and plotted against incubation time. A positive gradient of the best-fit regression line indicates *E. coli* growth whereas a negative gradient shows decay rate (*Lee et al., 2011*).

As protists are the major bacterial predators (*Enzinger & Cooper, 1976*), we also enumerated protists (*Caron, 1983*). A 50 mL water sample was preserved with glutaraldehyde (1% final concentration) during each sampling. At the laboratory, 1 to 2 mL preserved sample was filtered onto a black 0.8 µm polycarbonate filter (Millipore ATTP filter, Burlington, MA, USA) with a GF/A filter (Whatman, Little Chalfont, UK) as a backing filter. Filters were then rinsed twice with 0.1 M pH 4.0 Trizma-hydrochloride before being flooded with two mL of primulin solution (250 mg L$^{-1}$) for 15 mins. After staining, the solution was removed gently by vacuum filtration. The black filter was then placed on one drop of immersion oil on a clean glass slide, and the prepared slide was observed under an epifluorescence microscope (Olympus BX60F-3, Tokyo, Japan) with U-MWU filter cassette (excitor 330–385 nm, dichroic mirror 400 nm, barrier 420 nm).

## Habitat transition experiment for *E. coli*

We added 2 g of fresh sediment sample to the bottom of an autoclaved universal bottle (121 °C at 15 psi for 15 min), carefully avoiding any contact with the inner wall of the bottle. Then 0.4% (w/v) sterile soft agar (Difco, East Rutherford, NJ, USA) (kept at 45 °C) was added to the bottle until it covers approximately one cm level above the sediment. After the agar solidified, 18 mL of sterile saline was added slowly (Fig. 2). To check for contamination, a blank without addition of sediment sample was also carried out. The

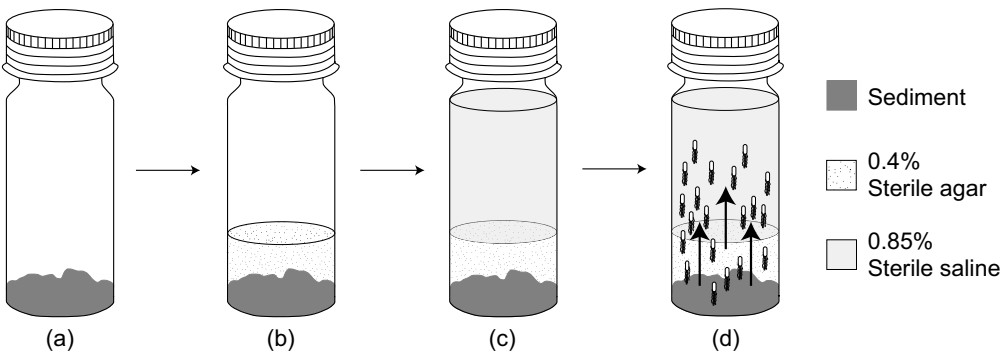

**Figure 2  A flow diagram of the habitat transition experiment.** (A) Addition of 2 g sediment sample. (B) Addition of 0.4% sterile agar on top of the sediment sample. (C) Addition of 18 mL 0.85% sterile saline after agar solidified. (D) Habitat transition of sediment *E. coli* (not to scale) to the water column during incubation.

habitat transition experiment was then incubated at 30 °C for 24 h in dark. The abundance of *E. coli* in the overlying saline was enumerated every 6 h *via* MFT on m-TEC agar. The cfu data was then natural logarithm transformed and plotted against incubation time where the gradient of the best-fit regression line was determined as *E. coli* increase rate ($\mu_{increase}$).

As *E. coli* also undergoes intrinsic growth in the saline environment (*Hrenović & Ivanković, 2009*), we setup a microcosm experiment by replacing raw sediment sample with autoclaved sediment sample from Tasik Kelana ($n = 2$) and Tasik Central Park Bandar Utama ($n = 2$). The sediment was autoclaved to replicate possible nutrient contribution from the sediment but prevent adding bacteria to the microcosm. We then inoculated $10^3$ cfu mL$^{-1}$ of *E. coli* to the sterile saline. The microcosm was then incubated at 30 °C for 24 h in the dark and the abundance of *E. coli* in the saline was enumerated every 6 h *via* MFT on m-TEC. The best-fit linear slope was determined as *E. coli* intrinsic growth ($\mu_{intrinsic}$), and the habitat transition rate was finally estimated by the following equation: $\mu_{increase} - \mu_{intrinsic}$.

## Data analysis

All data were reported in this study as mean ± standard deviation (SD) unless stated otherwise. Values beyond mean ± 2× SD were determined as outliers, and the coefficient of variation (*CV*) was used to measure the dispersion of data. Before statistical analysis, bacterial cfus were transformed by log (cfu + 1), whereas for growth or decay rate estimations, bacterial cfus were transformed *via* natural logarithm. Correlation analysis was carried out to identify relationships among variables, whereas linear regression was used for rate analysis. Student's *t*-test was used to compare between groups, whereas one-way ANOVA (analysis of variance) with Tukey's *post-hoc* analysis was used to determine the differences among lakes, and $p \leq 0.05$ was considered significant. PAST (PAleontological STatistics) software (version 4.09) for Windows (*Hammer, Harpper & Ryan, 2001*) was used to perform the statistical analyses, whereas plots were made in GraphPad Prism (version 9.5.1.733) for Windows (*Swift, 1997*).

**Table 1** The size and land use of five lakes, and water physico-chemical variables (Mean ± SD) measured at five lakes in this study.

| | Tasik Varsiti ($n = 4$) | Tasik Taman Jaya ($n = 3$) | Tasik Aman ($n = 4$) | Tasik Kelana ($n = 4$) | Tasik Central Park Bandar Utama ($n = 3$) |
|---|---|---|---|---|---|
| Area (m$^2$)/Land use | 13,102.74/educational | 28,804.52/residential and commercial | 17,778.91/residential | 67,074.41/residential and commercial | 1,786.64/commercial |
| Temperature (°C) | 29.9 ± 0.6 | 28.7 ± 0.8 | 29.6 ± 1.1 | 29.9 ± 1.2 | 29.9 ± 0.6 |
| pH | 7.2 ± 0.3 | 6.8 ± 0.4 | 7.2 ± 0.3 | 6.9 ± 0.3 | 7.2 ± 0.6 |
| DO (mg L$^{-1}$) | 9.01 ± 0.68 | 2.65 ± 1.78 | 9.47 ± 3.93 | 9.43 ± 4.39 | 9.65 ± 2.00 |
| TSS (mg L$^{-1}$)[***] | 26 ± 6[a] | 33 ± 6[b] | 31 ± 8[c] | 20 ± 7[d] | 65 ± 13[abcd] |
| POM (mg L$^{-1}$)[***] | 20 ± 1[a] | 13 ± 2[bc] | 25 ± 4[bde] | 14 ± 5[df] | 43 ± 5[acef] |
| Chl $a$ (μg L$^{-1}$)[***] | 50.89 ± 14.54[ab] | 31.03 ± 3.62[cd] | 90.63 ± 15.14[ace] | 55.04 ± 13.44[e] | 84.00 ± 2.60[bd] |
| NH$_4$ (μM)[***] | 0.43 ± 0.18[ab] | 79.36 ± 26.27[acde] | 1.67 ± 1.18[cf] | 37.54 ± 23.38[bdf] | 0.97 ± 0.25[e] |
| PO$_4$ (μM) | 0.18 ± 0.03 | 0.37 ± 0.04 | 0.34 ± 0.18 | 0.29 ± 0.16 | 0.21 ± 0.02 |

**Notes.**
[***] Showed significant differences in ANOVA at $p < 0.001$.
The same superscript letter of the alphabet indicates significant difference after Tukey's pairwise analysis.

## RESULTS

### Environmental variables

Table 1 lists the size and land use of five lakes, and physico-chemical variables measured in the water samples collected at the five lakes in this study. Surface water temperature and pH varied little among the five lakes and ranged from 28.7 ± 0.8 °C to 29.9 ± 1.2 °C (*CV* = 3%) and 6.8 ± 0.4 to 7.2 ± 0.3 (*CV* = 5%), respectively. In contrast, DO levels varied among the five lakes, from 2.65 ± 1.78 mg L$^{-1}$ at Tasik Taman Jaya to 9.65 ± 2.00 mg L$^{-1}$ at Tasik Central Park Bandar Utama (ANOVA: $n = 18$, $F(4,13) = 3.08$, $p = 0.05$).

In contrast, TSS and POM concentrations were different among the lakes, and ranged from 21 ± 7 mg L$^{-1}$ to 65 ± 13 mg L$^{-1}$ (ANOVA: $n = 18$, $F(4,13) = 14.06$, $p < 0.001$), and from 13 ± 2 mg L$^{-1}$ to 43 ± 5 mg L$^{-1}$ (ANOVA: $n = 18$, $F(4,13) = 34.8$, $p < 0.001$), respectively. TSS and POM concentrations were highest at Tasik Central Park Bandar Utama (Tukey's HSD: TSS: $q > 6.65$, $p < 0.01$; POM: $q > 6.41$, $p < 0.01$). Chl $a$ concentration also varied among lakes (ANOVA: $n = 18$, $F(4,13) = 14.14$, $p < 0.001$), and was highest at Tasik Aman (90.63 ± 15.14 μg L$^{-1}$) (Tukey's HSD: $q > 5.06$, $p < 0.03$). For dissolved inorganic nutrients, NH$_4$ varied among five lakes (ANOVA: $n = 18$, $F(4,13) = 16.85$, $p < 0.001$), ranged from 0.30 to 98.83 μM and was highest at Tasik Taman Jaya (Tukey's HSD: $q > 4.70$, $p < 0.04$), whereas PO$_4$ was similar among the lakes (ANOVA: $n = 18$, $F(4,13) = 1.80$, $p = 0.19$), and varied from 0.15 to 0.60 μM.

For the physico-chemical properties of sediments (Table 2), average particle size ranged from 55.2 ± 26.2 to 613.4 ± 124.2 μm, and were different among the five lakes (ANOVA: $n = 18$, $F(4,13) = 21.62$, $p < 0.001$) with the largest average particle size at Tasik Kelana (Tukey's HSD: $q > 5.52$, $p < 0.02$). The sediment texture at Tasik Varsiti was mainly loam, whereas in other lakes were mainly sand. Average sediment organic matter measured

**Table 2 Sediment composition, particle size and organic matter measured at five stations in this study.**

| Station | USDA Textural class[a] | Particle size (μm)[***] | Sediment composition (%) | | | Organic matter (mg g$^{-1}$) |
|---|---|---|---|---|---|---|
| | | | Sand | Clay | Silt | |
| Tasik Varsiti ($n = 4$) | Loam | $117.35 \pm 80.86^{ab}$ | 47.35% ± 18.41% | 14.43% ± 8.96% | 38.23% ± 9.75% | 30 ± 11 |
| Tasik Taman Jaya ($n = 3$) | Coarse sand | $340.47 \pm 99.57^{ac}$ | 87.08% ± 6.36% | 1.55% ± 2.68% | 11.38% ± 3.79% | 33 ± 10 |
| Tasik Aman ($n = 4$) | Sandy loam | $138.56 \pm 102.74^{d}$ | 64.15% ± 20.82% | 16.01% ± 15.80% | 19.84% ± 16.68% | 27 ± 14 |
| Tasik Kelana ($n = 4$) | Coarse sand | $627.14 \pm 107.96^{bcde}$ | 97.93% ± 1.98% | 0% | 2.07% ± 1.98% | 5 ± 2 |
| Tasik Central Park Bandar Utama ($n = 3$) | Loamy sand | $253.97 \pm 38.65^{e}$ | 82.2% ± 2.55% | 5.33% ± 1.32% | 12.47% ± 1.84% | 33 ± 26 |

Notes.
[***] Showed significant difference in ANOVA at $p < 0.001$.
The same superscript letter of the alphabet indicates significant differences after Tukey's pairwise analysis.
[a] Sediment textural class was determined according to the United States Department of Agriculture (*Soil Science Division Staff, 2017*).

ranged from 3 to 72 mg g$^{-1}$ and was not different among the five lakes (ANOVA: $n = 18$, $F_{(4,13)} = 2.96$, $p = 0.06$).

## Biotic variables

Total coliform and *E. coli* were detected in all five urban lakes (Fig. 3, Table S2). Total coliform ranged from 21 to 4,600 cfu mL$^{-1}$, and *E. coli* ranged from 1 to 2,300 cfu mL$^{-1}$. Total coliform in the water was different among the five lakes (ANOVA: $n = 20$, $F_{(4,15)} = 3.58$, $p = 0.03$) but *E. coli* count was not different (ANOVA: $n = 20$, $F_{(4,15)} = 2.52$, $p = 0.09$). The highest total coliform was detected at Tasik Kelana (Tukey's HSD: $q = 4.97$, $p = 0.02$). For the urban lake sediments, *E. coli* was present in all five lake sediments (Fig. 4). Its abundance ranged from below detection to 12,000 cfu g$^{-1}$, and there was no difference among the five lakes (ANOVA: $n = 20$, $F_{(4,15)} = 2.69$, $p = 0.07$).

### *E. coli* decay or growth rates

Generally, the abundance of *E. coli* in the larger fractions (total and <20 μm fraction) decreased with incubation time, while the <0.2 μm fraction increased (Figs. 5 and 6, Table S3). Decay rates among the five lakes in the total fraction (ANOVA: $n = 19$, $F_{(4,14)} = 7.85$, $p < 0.01$) and in the <20 μm fraction (ANOVA: $n = 18$, $F_{(4,13)} = 4.89$, $p = 0.01$) were different (Fig. 7, Table S4). The highest decay rates in both the total fraction (Tukey's HSD: $q > 4.65$, $p < 0.04$) and <20 μm fraction (Tukey's HSD: $q = 5.77$, $p = 0.01$) were observed at Tasik Taman Jaya. The decay rates measured in the total fraction also did not differ from those in the <20 μm fraction (Student's *t*-test: $n = 37$, $t_{(35)} = 0.43$, $p = 0.67$). As decay was most likely attributed to protistan grazers (*Lee et al., 2011*), we measured protists abundance in the water samples, and observed that protists counts ranged from $3.04 \times 10^4$ cells mL$^{-1}$ to $6.93 \times 10^4$ cells mL$^{-1}$ but showed no differences among the five lakes (ANOVA: $n = 15$, $F_{(4,10)} = 2.18$, $p = 0.14$). In contrast, *E. coli* grew in the <0.2 μm

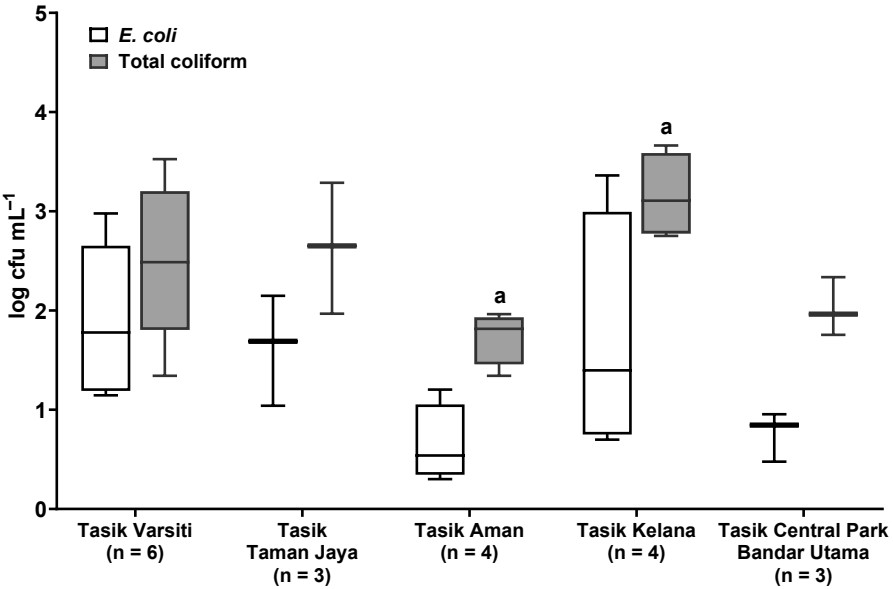

**Figure 3 Box-and-whisker plot showing the range and median of *E. coli* and total coliform (log cfu mL$^{-1}$) in the waters of the five stations in this study.** Whisker shows the minimal to maximal bacteria abundance and the box shows the interquartile range, while the horizontal line represents the median. The same letters of the alphabet are used to indicate significant differences after Tukey's pairwise analysis.

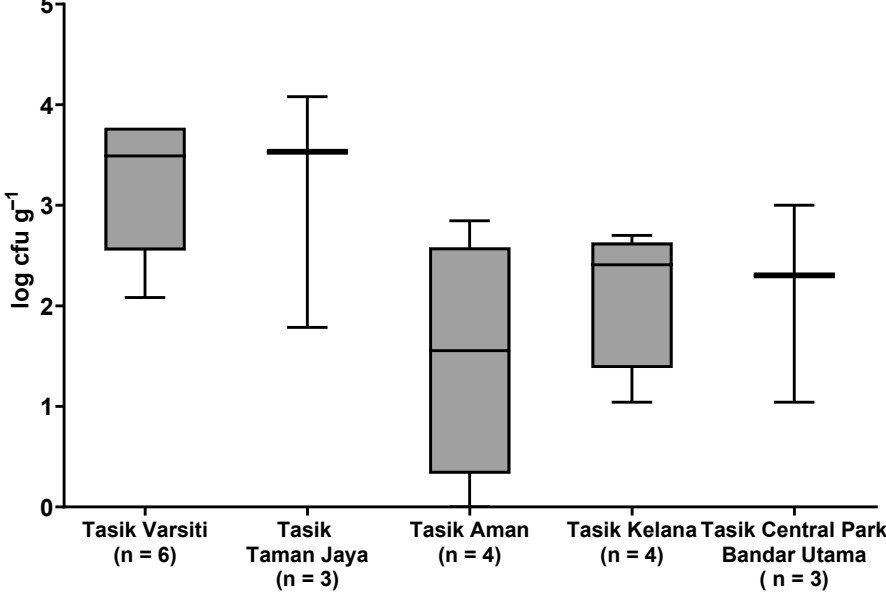

**Figure 4 Box-and-whisker plot showing the range and median of *E. coli* (log cfu g$^{-1}$) in the sediments of the five stations in this study.** Whisker shows the minimal to maximal bacteria abundance and the box shows the interquartile range, while the horizontal line represents the median.

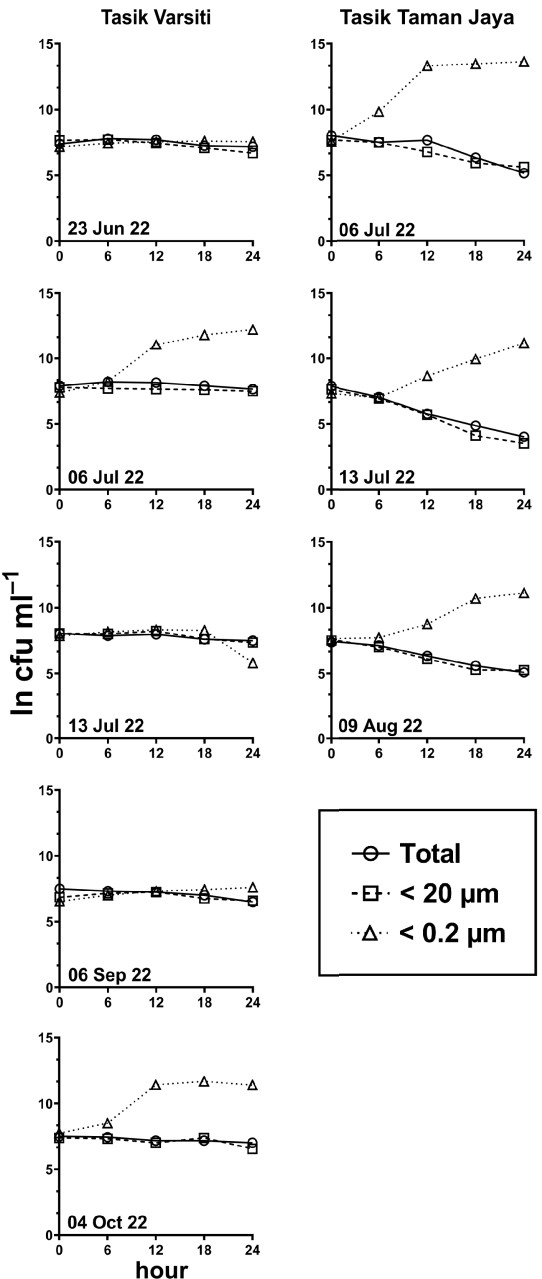

**Figure 5** *E. coli.* decay or growth over time (ln cfu mL$^{-1}$) in total, <20 μm and <0.2 μm fractions. Measured at Tasik Varsiti ($n = 5$), Tasik Taman Jaya ($n = 3$).

fraction, and the *E. coli* growth rates varied among five lakes (ANOVA: $n = 18$, $F$ (4,13) = 3.65, $p = 0.03$).

## Habitat transition experiment for *E. coli*

In the habitat transition experiment, *E. coli* abundance generally increased with time (Fig. 8, Table S5). *E. coli* increase rates ($\mu_{increase}$) in the water column ($p < 0.05$) ranged from 0.40

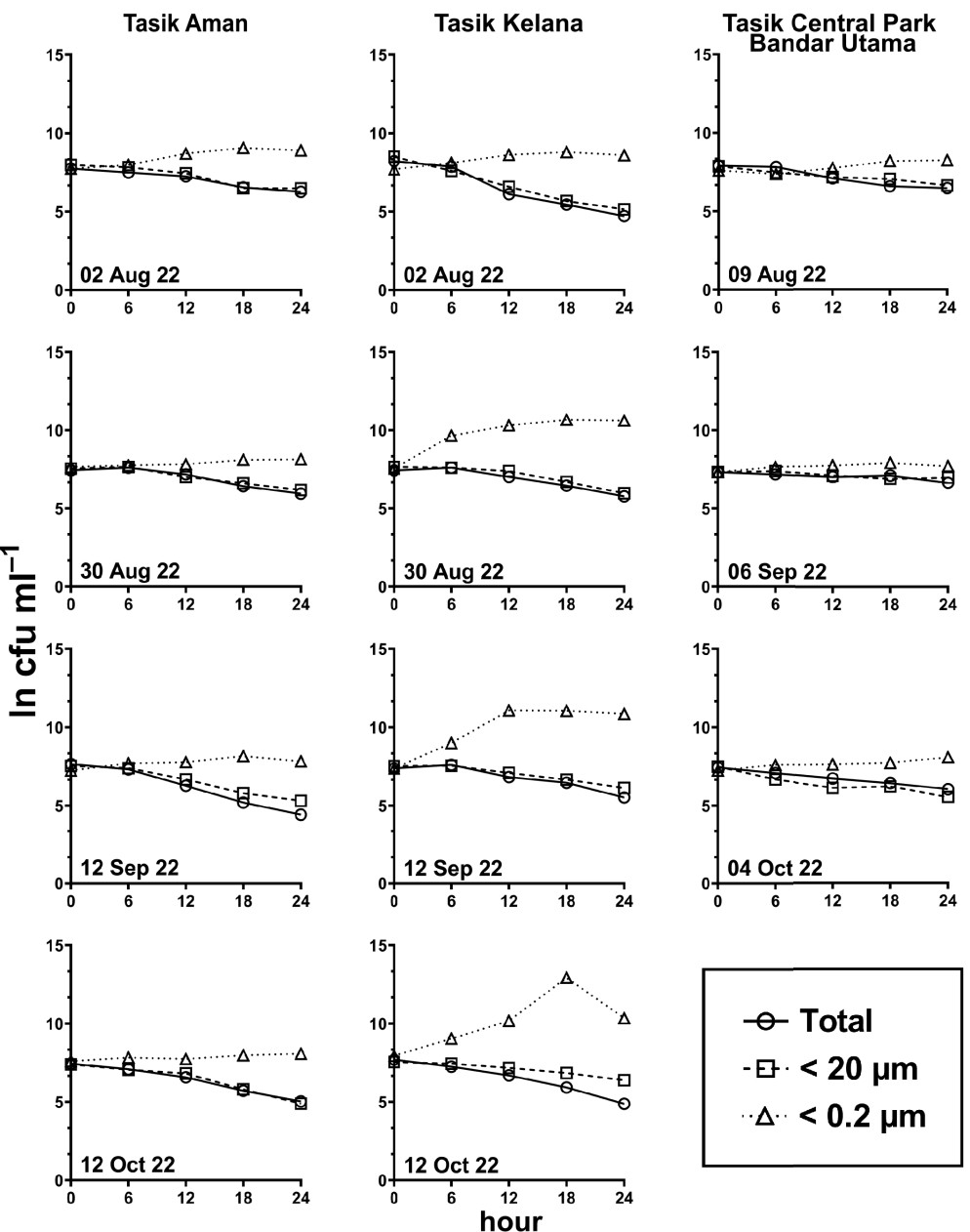

**Figure 6** *E. coli* **decay or growth over time (ln cfu mL$^{-1}$) in total, <20 μm and <0.2 μm fractions.** Measured at Tasik Aman ($n = 4$), Tasik Kelana ($n = 4$) and Tasik Central Park Bandar Utama ($n = 3$).

to 0.59 h$^{-1}$ at Tasik Varsiti, 0.46 to 0.62 h$^{-1}$ at Tasik Taman Jaya, 0.41 to 0.74 h$^{-1}$ at Tasik Aman, 0.61 to 0.71 h$^{-1}$ at Tasik Kelana and 0.69 to 0.78 h$^{-1}$ at Tasik Central Park Bandar Utama (Table S6).

As the *E. coli* increase rate is a sum of both transition and intrinsic growth, we also measured *E. coli* intrinsic growth rates (Table S7). The intrinsic growth rates using sterile sediments from Tasik Kelana were 0.39 h$^{-1}$ and 0.32 h$^{-1}$, and were similar to Tasik Central
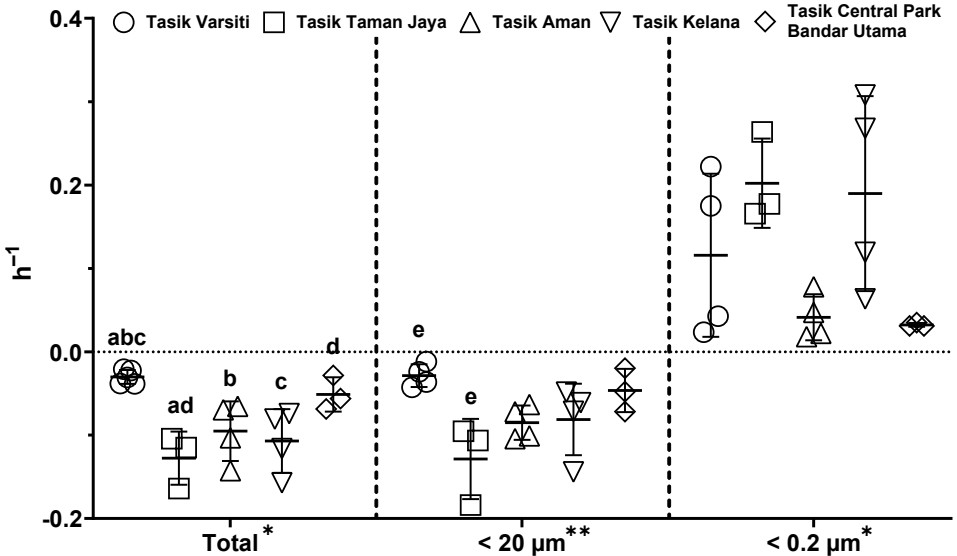

**Figure 7  Scatter dot-plots of *E. coli* decay and growth rates ($p \leq 0.05$) for each station.** Mean $\pm$ SD is represented by a plus symbol with error bars. One asterisk (*) and two asterisks (**) showed significant differences in ANOVA at $p < 0.05$ and $p < 0.01$, respectively, among stations in the fractions. Same letter of the alphabet indicates significant differences after Tukey's pairwise analysis.

Park Bandar Utama (0.41 h$^{-1}$ and 0.36 h$^{-1}$). Although sediments at Tasik Kelana had the lowest organic matter content (3 to 8 mg g$^{-1}$), whereas Tasik Central Park Bandar Utama had the highest (19 to 72 mg g$^{-1}$) among the five lakes, their *E. coli* intrinsic growth rates were not different (ANOVA: $n = 4$, $F(1,2) = 0.48$, $p = 0.56$). Therefore, for the calculation of habitat transition rates, we assumed the average intrinsic growth rate ($0.37 \pm 0.04$ h$^{-1}$) for all five lakes (Fig. 9, Table S8). The habitat transition rates were different among five lakes (ANOVA: $n = 18$, $F(4,13) = 4.01$, $p = 0.02$), with the highest at Tasik Central Park Bandar Utama (Tukey's HSD: $q = 4.67$, $p = 0.04$).

# DISCUSSION

## Environmental condition of the urban lakes

The surface water temperatures recorded at the five lakes were relatively high with low variability, and is typical of tropical waters (*Lim et al., 2018*). The DO concentrations measured at Tasik Varsiti, Tasik Aman, Tasik Kelana and Tasik Central Park Bandar Utama were at healthy levels, and within the range previously reported for tropical freshwater (*Wong et al., 2022*). However, for Tasik Taman Jaya, we observed hypoxic levels ($2.65 \pm 1.78$ mg L$^{-1}$) (*Farrell & Richards, 2009*), which was not surprising as *Wong et al. (2022)* had previously classified Tasik Taman Jaya at Class III for total coliform and Class V for faecal coliform, indicative of extensive treatment required for the suitability of water supply (*Department of Environment, 2008*). All lakes were also observed with high Chl *a*, indicating varying levels of eutrophication (*Lim et al., 2018*).
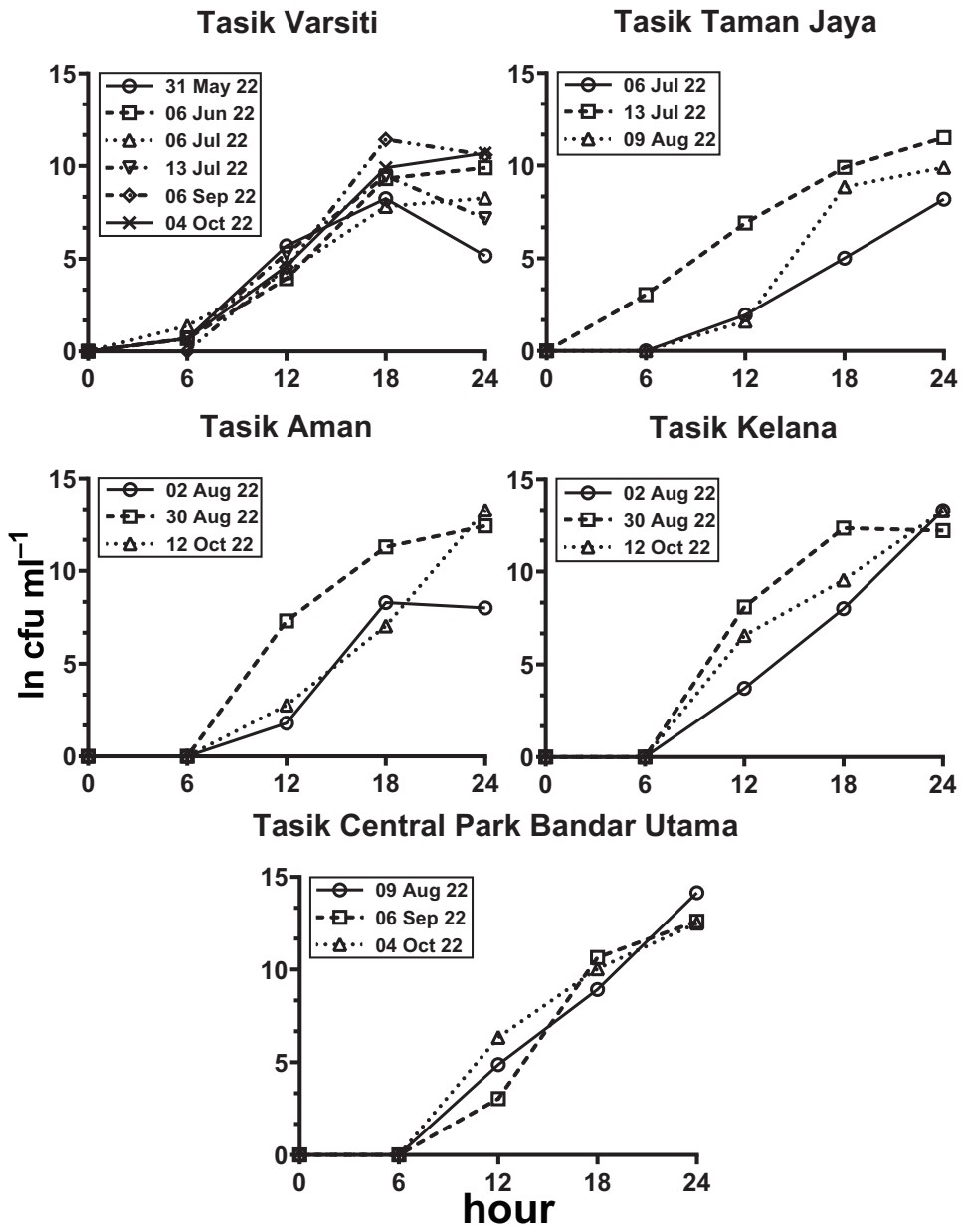

**Figure 8** **Increase in *E. coli* abundance into the overlying water (ln cfu mL$^{-1}$) against incubation time.** Measured with sediments from Tasik Varsiti ($n = 6$), Tasik Taman Jaya ($n = 3$), Tasik Aman ($n = 3$), Tasik Kelana ($n = 3$) and Tasik Central Park Bandar Utama ($n = 3$).

## Total coliform and *E. coli* in water and sediment of urban lakes

The total coliform abundance enumerated in five lake waters were within the range previously reported in Malaysia (*Wong et al., 2022*). According to National Water Quality Standards for Malaysia, Tasik Varsiti, Tasik Taman Jaya and Tasik Kelana were categorized as Class V for total coliform, whereas Tasik Aman and Tasik Central Park Bandar Utama

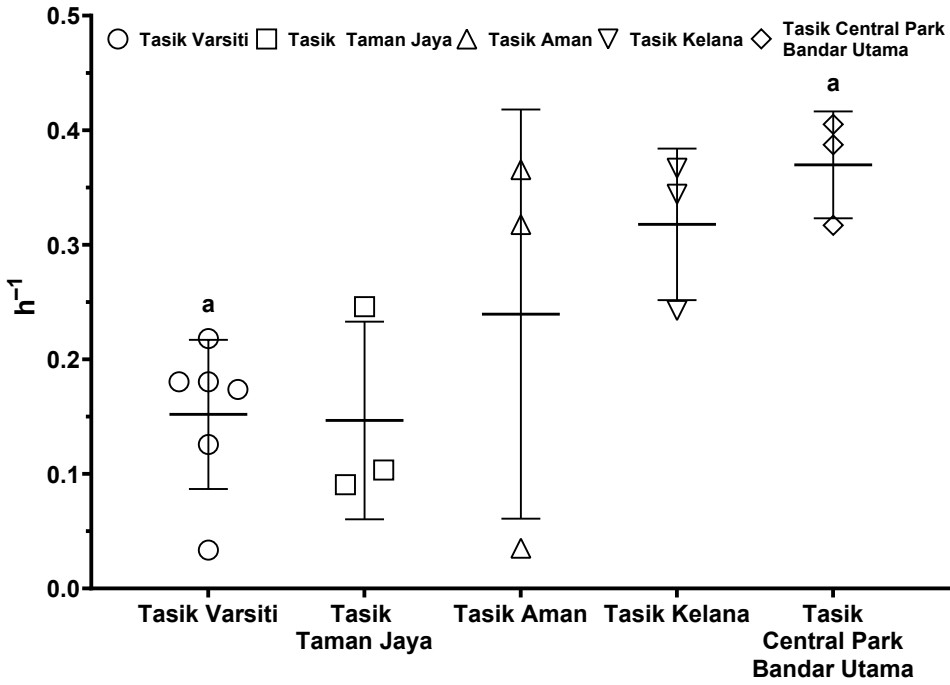

**Figure 9 Scatter dot-plots of *E. coli* habitat transition rates ($p < 0.05$) for each station.** Mean $\pm$ SD is represented by a plus symbol with error bars. Same letters indicate significant difference after Tukey's pairwise analysis.

as Class III. Relative to *Wong et al. (2022)*, the water quality in these lakes had deteriorated over the last two years.

In our study, the abundance of *E. coli* in sediment and water were correlated (Pearson correlation: $n = 40$, $r$ (38) $= 0.53$, $p = 0.02$) (Fig. 10), similar to *Whitman, Nevers & Byappanahalli (2006)* and *Fluke, González-Pinzón & Thomson (2019)*. Although we did not use the same agar medium to enumerate *E. coli* abundance in water and sediment, we had previously shown that *E. coli* counts on m-TEC and CHROMagar ECC agar were strongly correlated (Regression: $n = 18$, $R^2 = 0.99$, $F$ (1,7) $= 561.35$, $p < 0.001$) (Fig. 11, Table S9), and that the abundance of *E. coli* (log cfu mL$^{-1}$) obtained on m-TEC was on average 149% higher than that on CHROMagar ECC. In order to compare *E. coli* abundance in water and sediment, we corrected the abundance obtained, and found that the abundance of *E. coli* in the sediment was still higher than in the water column (*Stephenson & Rychert, 1982*; *An, Kampbell & Peter Breidenbach, 2002*; *Garzio-Hadzick et al., 2010*; *Pandey et al., 2018*; *Fluke, González-Pinzón & Thomson, 2019*).

Relative to the water column, sediment provides *E. coli* with more nutrients (*Jamieson et al., 2005*); lower UV intensity (*Koirala et al., 2008*); lesser bacterivore grazing (*Wright et al., 1995*) and lower oxygen (*Lorke & MacIntyre, 2009*). This host intestinal-like environment helps *E. coli* to survive better in sediments, even at varying climates (*Ishii et al., 2006*; *Ishii & Sadowsky, 2008*; *Garzio-Hadzick et al., 2010*; *Rochelle-Newall et al., 2015*; *Tymensen et al., 2015*; *Fluke, González-Pinzón & Thomson, 2019*).

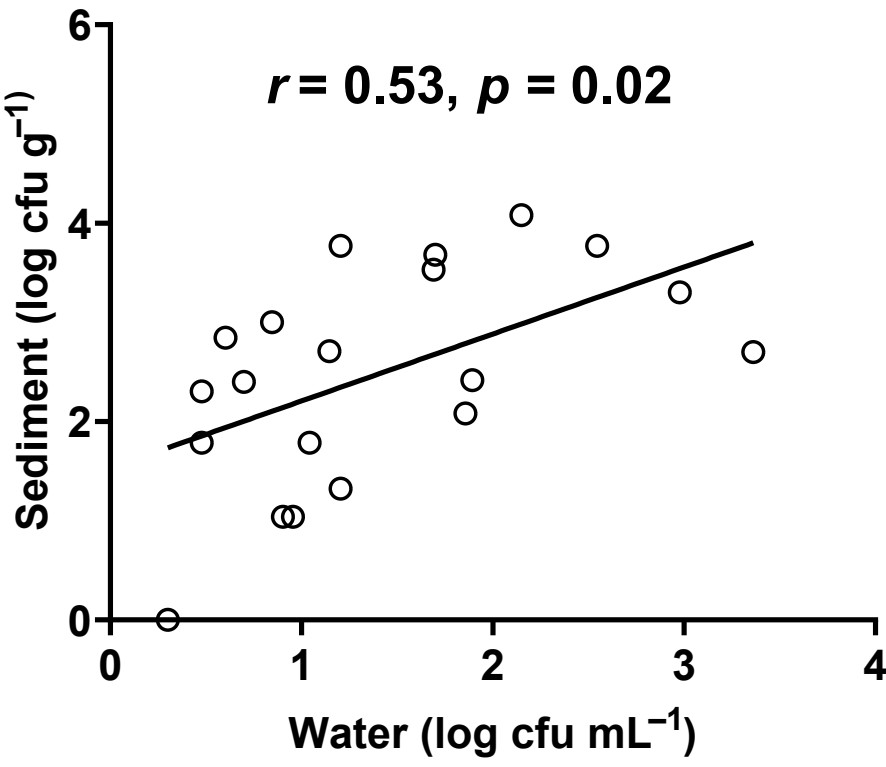

**Figure 10** Correlation between the abundance of *E. coli* in water and sediment.

As *E. coli* is dependent upon sediment organic matter for growth (*Ishii et al., 2006*), higher abundance of *E. coli* in sediments with higher organic matter has been reported (*Lee et al., 2006*). However in our study, sediment organic matter and sediment *E. coli* abundance were not correlated (Pearson correlation: $n = 36$, $r$ (34) $= -0.11$, $p = 0.66$). One possible reason could be the organic matter replete state in our lakes. The sediment organic matter from this study was relatively higher (Organic matter content: Tasik Varsiti 3%, Tasik Taman Jaya 3.3%, Tasik Aman 2.7%, Tasik Kelana 0.5% and Tasik Central Park Bandar Utama 3.3%), than that reported by *Lee et al. (2006)* (*i.e.,* 0.7%–1.1%). In this study, the sediment *E. coli* abundance also did not correlate with particle size (Pearson correlation: $n = 36$, $r$ (34) $= -0.07$, $p = 0.78$), further substantiating that sediment texture may be less important for the survival of sediment *E. coli* (*Lee et al., 2006*).

Although sediments could act as reservoirs of *E. coli*, and contribute to the water column *E. coli*, *E. coli* dynamics in the water column is also dependent upon *E. coli* decay or growth rates in the water (*Lee et al., 2011*). The decay rate in total and <20 μm fractions were higher than the smaller fraction and consistent with previous reports (*Lee et al., 2011*; *Wong et al., 2022*). In the total fraction, the *E. coli* decay rates ranged from 0.02 to 0.16 h$^{-1}$ (or 0.50 to 3.94 d$^{-1}$). These decay rates were generally within the range reported by *Flint (1987)* measured at 37 °C, and higher than in subtropical water (*Bitton et al., 1983*). Temperature

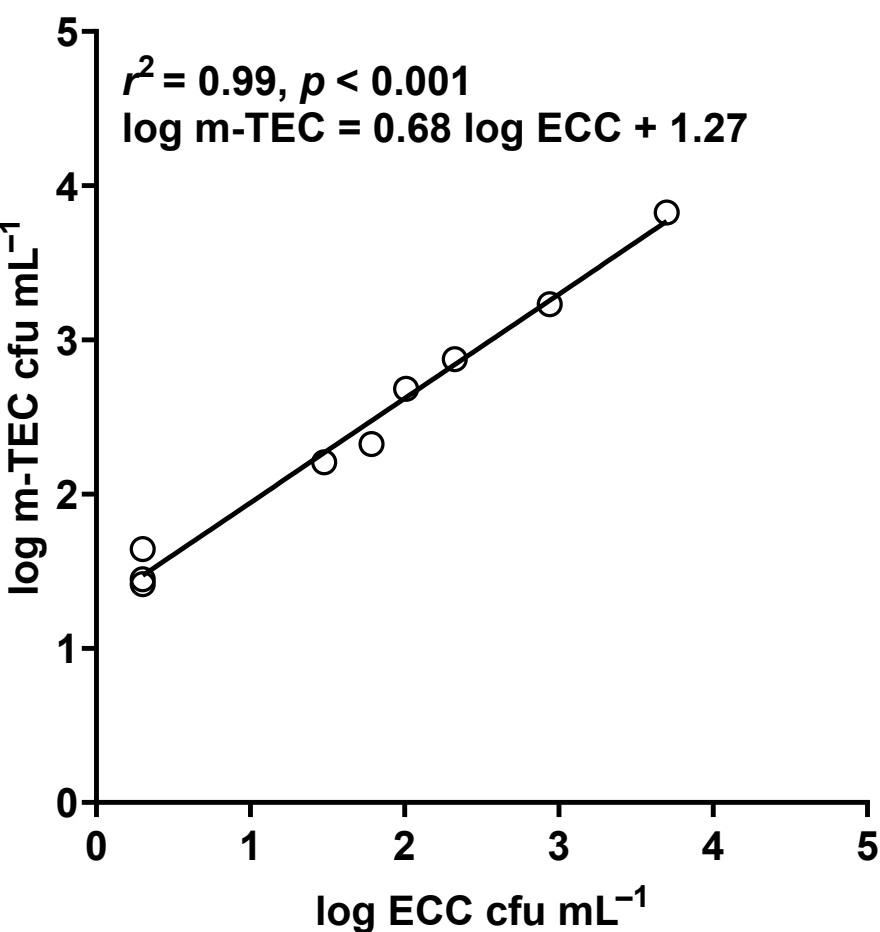

**Figure 11  Correlation between log m-TEC cfu mL$^{-1}$ and log ECC cfu mL$^{-1}$ in the enumeration of *E. coli* in water.**  m-TEC stands for m-TEC agar, ECC stands for CHROMagar ECC agar.

may explain the generally higher rates observed in this study, as microbial activity is at its optimum in tropical aquatic habitats (*White et al., 1991*).

In our study, the decay of *E. coli* in the larger fraction is mainly due to protistan bacterivory (*Enzinger & Cooper, 1976*; *Lee et al., 2011*; *Wong et al., 2022*). However, we found no correlation between decay rate and protist counts (Pearson correlation: $n = 10$, $r(8) = -0.3$, $p = 0.62$). As *E. coli* only accounts for small fraction of the total bacterial community, at about 4.48% of total culturable gram-negative rod in freshwater (*Goñi Urriza et al., 1999*), this could explain the uncoupling between protists and *E. coli* decay rates. *Lee et al. (2011)* have also reported that *E. coli* decay rates have a relatively small impact on the overall bacterivory rate. Although viral lysis could also cause *E. coli* mortality, previous studies have shown that its role is generally minimal (*Lee et al., 2011*). Moreover in the <0.2 μm fraction, where protists were removed, *E. coli* did not decrease but increased against incubation time, suggesting that viral lysis was not significant (*Lee et al., 2011*).

### The role of habitat transition rates for *E. coli* persistence in the water column

Habitat transition experiments in this study showed that *E. coli* in sediments could transition from the sediment to the overlying water column without mechanical effects *i.e.,* turbulence and resuspension. Although seasonal change in precipitation and turbulence can cause sediment resuspension and an increase in *E. coli* abundance in the upper water column (*Li, Filippelli & Wang, 2023*), the effect should be minimal in lakes as *E. coli* abundance quickly return to pre-resuspension level (*Whitman, Nevers & Byappanahalli, 2006*; *Abia et al., 2017*). Moreover, we have shown net transition rates in laboratory experiments without mechanical effects. Therefore, any precipitation will only increase the impact of habitat transition, and not affect the conclusion from this study.

The habitat transition rates fluctuated among lakes ($CV = 53\%$) and was not correlated with sediment particle size (Pearson correlation: $n = 32$, $r$ $(30) = 0.38$, $p = 0.14$) and organic matter (Pearson correlation: $n = 32$, $r$ $(30) = -0.08$, $p = 0.78$). Although the habitat transition of *E. coli* from sediment to water may be associated with biofilm sloughing (*Lee et al., 2006*), the mechanisms that drive dispersal in *E. coli* biofilms are complicated, and some are still unknown (*McDougald et al., 2012*).

In this study, we observed the presence of *E. coli* in all five tropical urban lake waters. Although previous studies have also reported on the survival of *E. coli* in the water column, they did not include the effects of sediment (*Lee et al., 2011*; *Wong et al., 2022*). Given the higher abundance of *E. coli* in the sediments (*Garzio-Hadzick et al., 2010*; *Fluke, González-Pinzón & Thomson, 2019*), and that these *E. coli* can transition to the water column, the effects of sediment on the abundance of *E. coli* in the water column could be important.

In order to evaluate the effect of habitat transition on the abundance of *E. coli* in the water column, we compared *E. coli* habitat transition rates with total fraction decay rates. We found that in most cases (>80%), the habitat transition rates were higher than the total fraction decay rates (Fig. 12). Thus, there was a net increase rate of *E. coli* in the water column, calculated by the following equation: $\mu_{\text{habitat transition}} - \mu_{\text{total fraction decay}}$. The *E. coli* net increase rates ranged up to 0.36 $\text{h}^{-1}$ (0.16 $\pm$ 0.13 $\text{h}^{-1}$) in our study. When the habitat transition rate exceeds the total fraction decay rate, using *E. coli* as a faecal indicator could overestimate the faecal contamination level.

However, these rates were from microcosm-based experiments that did not include other biotic (*e.g.*, biofilm and competition) and abiotic (sunlight) factors that can also affect *E. coli* survivability *in-situ* (*Korajkic et al., 2014*; *Stocker et al., 2019*; *Petersen & Hubbart, 2020*; *Moon et al., 2023*). Further studies are therefore needed to understand the role of these factors on *E. coli* habitat transitions. In this study, we showed the role of sediment as reservoirs and habitat transition as a possible explanation for the persistence of *E. coli* in tropical aquatic habitats (*Wong et al., 2022*).

## CONCLUSIONS

Sediments acted as a reservoir of *E. coli* in tropical urban lakes with a higher abundance of *E. coli* than in the water column. The habitat transition of *E. coli* from sediment to the
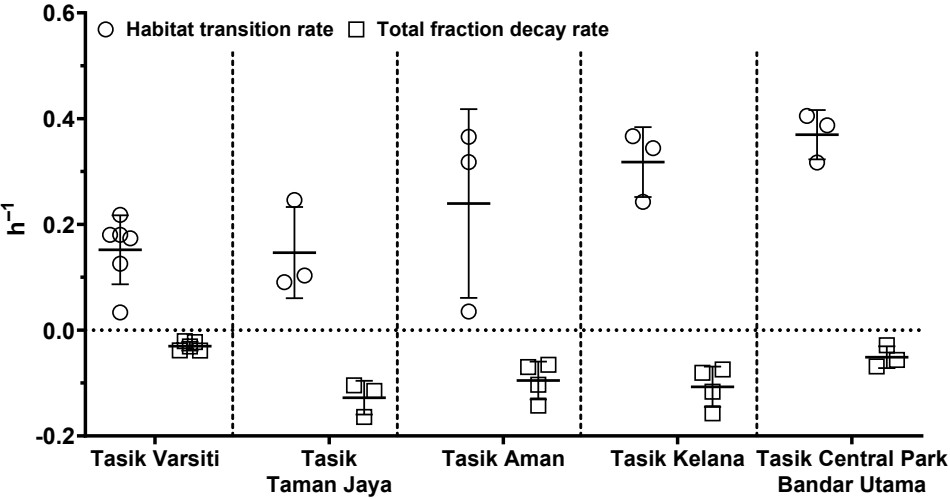

**Figure 12 Scatter dots plot of habitat transition and total fraction decay rates of _E. coli_ in water column of five stations measured in this study.** Mean ± SD is represented by a plus symbol with error bars.

water column affects its abundance in the water column, and could be one of the reasons for the persistence of _E. coli_ in tropical urban lakes.

# ACKNOWLEDGEMENTS

We thank Yi You Wong, Kyle Young Low, Walter Aaron and Ee Lean Thiang for their assistance with sampling.

## Funding

This work was supported by the National Natural Science Foundation of China (Contract No: 41961144022) and the Ministry of Higher Education Malaysia for the HiCoE grant Phase II fund (MOHE-HiCoE IOES-2023C). The funders had no role in study design, data collection and analysis, decision to publish, or preparation of the manuscript.

## Grant Disclosures

The following grant information was disclosed by the authors:
National Natural Science Foundation of China: 41961144022.
Ministry of Higher Education Malaysia for the HiCoE: MOHE-HiCoE IOES-2023C.

## Competing Interests

The authors declare there are no competing interests.

## Author Contributions

- Boyu Liu conceived and designed the experiments, performed the experiments, analyzed the data, prepared figures and/or tables, authored or reviewed drafts of the article, and approved the final draft.

- Choon Weng Lee conceived and designed the experiments, performed the experiments, authored or reviewed drafts of the article, and approved the final draft.
- Chui Wei Bong conceived and designed the experiments, prepared figures and/or tables, and approved the final draft.
- Ai-Jun Wang analyzed the data, authored or reviewed drafts of the article, and approved the final draft.

### Data Availability

The raw data are available in the Supplemental Files.

### Supplemental Information

Supplemental information for this article can be found online at http://dx.doi.org/10.7717/peerj.16556#supplemental-information.

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
