# Peer review of "Investigating Escherichia coli habitat transition from sediments to water in tropical urban lakes"

_PeerJ, doi:10.7717/peerj.16556_

## Round 0.1 · original submission · Major Revisions

Please address all the reviewers' comments carefully and thoroughly. When re-submitting your manuscript, please also provide a point-by-point response letter.

Reviewer 1 ·

Basic reporting

Please add a paragraph on what factors were previously considered in influencing E.coli or bacteria habitat transition between sediments and water.

Line 53-59. The references used are rather old. Please add recent references if possible.
Line 53-59. I believe seasonal variation also affects the survival and abundance of E.coli in freshwater and aquatic habitat. I suggest adding more information about them as well.
Line 69-70. Please choose only important references; this is an exhaustive list.

Experimental design

Line 89-96. Please add more information about the lakes, such as depth, area (size), land use, proximity to each other, and whether they are independent lakes. Why was the sample taken every month at random dates? Certain lakes were missed for July sampling; please explain why four samples were taken in some lakes and three in others. There is mention of 35 water samples, but from figures 3 and 4, "n" values indicate 20. Was the E.coli abundance calculated for sediments as well? It is hard to see the data for 20 sediment samples. Please fix this. Precipitation is not taken into account since that impacts the mixing of sediment and water, causing the transfer of E.coli to the water environment.

Line 97-111. Total nitrogen and total phosphorous were not studied. Nutrient pollution is a significant contributor to E.coli survival. Please add that data if possible.

Line 112-119. Sediment nitrogen and phosphorous were not calculated; please provide a reason.

Validity of the findings

The two sections need to be re-written; please find my suggestions below,

The entire correlation data is not required; please find two examples below,
"Pearson correlation: n = 40, r(38) = 0.53, p = 0.02", “ANOVA: n = 18, F(4,13) = 14.06, p < 0.001” just mention whether the data was significant.

Line 228. It would be ideal if they were compared among lakes every month rather than taking an average. Data every month will illustrate whether temporal variations play a role in the abundance of E.coli.

Reviewer 2 ·

Basic reporting

The manuscript is well written, clearly communicates the study motivation and objectives, and cites relevant literature. Figures and tables are relevant and support main findings.

In the Results, findings are reported as differences among stations. I would recommend referring to them as lakes because it more clearly communicates the analyses are comparing whole lakes within a region. Stations could be interpreted as multiple locations within a single lake.

Figures 3, 7, 9 and Tables 1 &2
The more conventional method to report Tukey’s pairwise comparisons among groups is by labeling groups with the same letter that are NOT significantly different from one another. But the way the results are reported where letters depict significantly different groups in the MS may be alright because the authors explain how to interpret the letters. I leave it up to the editors and authors if they want to make any changes.

Figure 11
It would be helpful to define/distinguish TEC and ECC in the figure caption. Maybe something like “ECC (water) and TEC (sediment)”

Experimental design

The background, knowledge gaps, and motivation for the study are clearly communicated (L60-77). The methods are described in sufficient detail that they could be adopted by others and data used in the analyses are available in the supporting information files.

The study was conducted from samples collected from May – November. I was wondering if there are seasonal patterns that these lakes may experience that might affect E. coli transition rates? The sample period likely minimizes any seasonal variation, but it might be something to mention in the discussion whether seasonal patterns might affect E. coli abundances and rates.

Validity of the findings

The analyses and interpretation of the results are sound and could be replicated with the information provided. The main conclusions are clearly communicated. I really appreciate the Conclusion section of the manuscript that distills the detailed analyses into meaningful nuggets.

I couldn’t find descriptions of the supporting information files. Documentation/metadata of the supporting files would be helpful.

Additional comments

Overall, this study makes a useful contribution to environmental science/public health. The scope of the study aligns with PeerJ evaluation criteria. Below are general comments to improve clarity.

Reviewer 3 ·

Basic reporting

The aims of the article were clear. The article structure, figures, tables were reliable.

Experimental design

Research question well defined. The experiment methods described with sufficient detail & information to replicate.

Validity of the findings

The research results are unreliable.

Additional comments

The paper has conducted a lot of experiments in the water and sediment at five urban lakes in the Kuala Lumpur-Petaling Jaya area, but the innovation was not enough, and there are still some problems.
1. It is very thoughtful to question the faecal indicator bacteria (FIB), but relying on the water bodies and relevant experimental results in the paper, the experimentals is not enough to support this question” Therefore, a re-evaluation of E. coli as a faecal indicator bacterium in water surveillance and monitoring plan is needed especially in water bodies containing sediments. (Line 364-366)”. And the method used in the paper is relatively outdated (Line321-324, Flint 1987; Bitton et al., 1983; White et al., 1991). Therefore, the conclusion of this paper is inappropriate.
2. Maybe Escherichia coli has certain limitations, however, the paper did not propose a more reasonable method in the end. However, “Escherichia coli is a commonly used faecal indicator bacterium to assess the level of faecal contamination in aquatic habitats for many years (USEPA, 1986)” (Line 47+53). That is to say, there is currently no better replacement method, and simply raising doubts is not enough.
3. “From the results of the study, the habitat transition rates were higher than the decay rates in most cases (>80%)”(Line 39). And from Line 359-360: “The E. coli net increase rates ranged up to 0.36 h.1 (0.16 ± 0.13 h.1) in our study.” If this result was correct, what will be the total number of bacteria in the water body after one year or three years? Is the conclusion reliable?

---

## Round 0.2 · Major Revisions

Please address the comments by the Reviewer 3 carefully and thoroughly. Make sure that all the comments are answered in a point-by-point response. We look forward to receiving your revised manuscript soon.

Reviewer 1 ·

Basic reporting

no comment

Experimental design

no comment.

Validity of the findings

no comment.

Reviewer 2 ·

Basic reporting

I am satisfied with the revisions the authors made.

Experimental design

I am satisfied with the revisions the authors made.

Validity of the findings

I am satisfied with the revisions the authors made.

Additional comments

none

Reviewer 3 ·

Basic reporting

The aims of the article were not clear. The article structure, figures, tables were reliable, but not necessary.

Experimental design

Research question didn’t well defined. The experiment methods described with sufficient detail & information to replicate.

Validity of the findings

The research results are unreliable.

Additional comments

The innovation was still not enough.
1. Line 90-95. “The absence of mechanical effects in the lake waters will help clarify the role of E. coli habitat transition.……” From this sentence, the aims and innovation were not clear. Furthermore, there are two literatures in the last sentence. That is to say, the description of this paper is not explained.
2. Line 326-327: “In this study, the sediment E. coli abundance also did not correlate with particle size”. Almost all of the sediment are particles except for the water, E. coli is just only a small part. Of course, the E. coli abundance also did not correlate with particle size. Here, the correlation of E. coli abundance to the corresponding particle of the sediment is reasonable.
3. Line 334-336: “In the total fraction, the E. coli decay rates ranged from 0.02 to 0.16 h-1 (or 0.50 to 3.94 d-1). These decay rates were generally within the range reported by Flint (1987) measured at 37℃, and higher than in subtropical water (Bitton et al., 1983). ” In 1987 and 1983, the decay rates of E. coli was clear, what’s the aims of this experiments?
4. Line 390-393: The conclusion is just a fact. This conclusion has no relation with this study.

Annotated reviews are not available for download in order to protect the identity of reviewers who chose to remain anonymous.

---

## Round 0.3 · accepted · Accept

The manuscript has been well revised and all the reviewers' comments have been addressed with satisfaction.